# Skin Function Improvement and Anti-Inflammatory Effects of Goat Meat Extract

**DOI:** 10.3390/foods13233934

**Published:** 2024-12-05

**Authors:** In-Seon Bae, Van-Ba Hoa, Jeong-Ah Lee, Won-Seo Park, Dong-Gyun Kim, Hyoun-Wook Kim, Pil-Nam Seong, Jun-Sang Ham

**Affiliations:** Animal Products Utilization Division, National Institute of Animal Science, RDA, Wanju 55365, Republic of Korea; b1983@rda.go.kr (V.-B.H.); 2970703@naver.com (J.-A.L.); pwonseo@rda.go.kr (W.-S.P.); kdg6589@rda.go.kr (D.-G.K.); woogi78@rda.go.kr (H.-W.K.); spn2002@rda.go.kr (P.-N.S.); hamjs@rda.go.kr (J.-S.H.)

**Keywords:** goat meat, HaCaT cells, RAW264.7 cells, anti-inflammatory, skin

## Abstract

Chronic skin conditions, such as atopic dermatitis, are characterized by a weakened skin barrier and persistent inflammation. Traditional treatments can frequently cause substantial side effects, emphasizing the need for safer alternatives. This study investigated the anti-inflammatory properties of goat meat extract and its effects on improving skin function. We conducted wound healing assays using HaCaT cells and analyzed the expression of key skin barrier-related genes. Additionally, the anti-inflammatory effects of goat meat extract were assessed in HaCaT cells stimulated with TNFα and IFNγ, as well as in LPS-treated RAW264.7 cells. Mechanistic studies focused on the activation of mitogen-activated protein kinase (MAPK) pathways. The results showed that goat meat extract significantly promoted wound closure in HaCaT cells and upregulated the expression of filaggrin, loricrin, and involucrin. The extract also reduced the production of pro-inflammatory cytokines and chemokines in both HaCaT and RAW264.7 cells. Furthermore, it inhibited the activation of the JNK, p38, and ERK pathways in TNFα/IFNγ-stimulated HaCaT cells. These findings suggest that goat meat extract improves skin barrier function and exhibits anti-inflammatory effects, indicating its potential as a therapeutic agent for chronic skin. Further research is required to investigate the in vivo effects of goat meat extract and validate its therapeutic potential.

## 1. Introduction

Atopic dermatitis (AD), along with other long-term inflammatory skin disorders, such as psoriasis and eczema, is marked by a compromised skin barrier and ongoing inflammation [1,2]. These conditions are prevalent worldwide, affecting millions of individuals, and result in significant discomfort, itching, and poor quality of life. The development of these diseases is influenced by a complex interaction of genetic, environmental, and immunological factors, along with abnormalities in skin barrier proteins and increased susceptibility to infections and allergens [1,3]. Skin barrier function is primarily maintained by structural proteins, such as filaggrin, involucrin, and loricrin, and tight junction proteins, including occludin and claudin [4,5]. These proteins play crucial roles in maintaining skin integrity and preventing transepidermal water loss. Pro-inflammatory cytokines, including TNFα, IL-1, and IL-6, are known to impair these proteins, leading to increased skin barrier dysfunction and inflammation [6,7]. Therefore, an agent that enhances barrier protein expression while simultaneously decreasing inflammatory cytokine production would be highly effective in managing skin conditions like AD [8,9,10].

Current treatments for AD and similar conditions often involve the use of topical corticosteroids, calcineurin inhibitors, or systemic immunosuppressants [11,12,13]. Although these treatments can be effective in managing symptoms, they are associated with significant side effects, including skin atrophy, increased risk of infection, and potential systemic toxicity. Consequently, exploring alternative and complementary therapies that can offer safer and more sustainable management of these skin conditions has attracted attention.

Recently, natural extracts have gained considerable attention owing to their potential therapeutic benefits in dermatological conditions. Plant-derived extracts, valued for their ability to moisturize and reduce inflammation, have been extensively studied. Ingredients such as aloe vera, chamomile, *Rorippa cantoniensis* Ohwi, and green tea extracts have shown promising results in enhancing skin hydration and reducing inflammation [10,14,15,16,17,18]. In contrast, animal-derived extracts have received less attention despite their potential benefits. Compounds such as snail mucin and fish collagen exhibit moisturizing and wound-healing properties [19,20,21,22]. However, the broader potential of products derived from other animals remains underexplored.

Goat meat is particularly notable for its nutritional benefits and its therapeutic properties. It has lower cholesterol and fat levels compared to lamb and beef, fewer calories than both beef and chicken, and is high in protein [23]. Additionally, goat meat is rich in unsaturated fatty acids, minerals, and amino acids, which enhance its digestibility and hypoallergenic properties, making it an ideal choice for children and individuals on restrictive diets [24,25,26,27]. Moreover, goat meat is a valuable source of vitamin B, pantothenic acid, folic acid, para-aminobenzoic acid, and choline. It also surpasses other livestock in concentrations of vitamins A, B1, and B2, which are known to promote skin hydration and strengthen skin immunity.

The health benefits of goat meat are further highlighted by its bioactive compounds, which exhibit anti-inflammatory effects and enhance skin barrier function [28,29]. Recent studies have emphasized the potential therapeutic benefits of goat meat extract in diverse applications. For example, supplementation with goat meat extract has been demonstrated to enhance exercise performance, alleviate physiological fatigue, and influence gut microbiota in mice [30]. This study aims to explore the effects of goat meat extract as a natural agent against AD.

## 2. Materials and Methods

### 2.1. Goat Meat Extract Preparation

Forty grams of goat meat were combined with 200 mL of distilled water. The resulting mixture was subjected to centrifugation at 1400× *g* for 30 min, after which the supernatant was collected. This supernatant was then heated for 2 h. Following this, it was filtered through a No. 1 filter paper. The obtained goat meat extract was freeze-dried and utilized in subsequent experiments.

### 2.2. Cell Culture

RAW264.7 (ATCC^®^ TIB-71^TM^, ATCC, Rockville, MD, USA) and HaCaT (CLS Cell Lines Service GmbH, Eppelheim, Germany) cells were grown in Dulbecco’s modified Eagle’s medium (DMEM; Welgene, Daegu, Republic of Korea), enriched with 10% fetal bovine serum (FBS; Sigma-Aldrich, St. Louis, MO, USA) and 1% penicillin–streptomycin (Sigma-Aldrich). The cells were incubated at 37 °C in a humidified incubator containing 5% CO_2_ atmosphere. This study utilized untreated HaCaT cells as the control group. HaCaT cells were exposed to goat meat extract at concentrations of 50, 100, and 200 µg/mL. To induce inflammation in RAW264.7 cells, they were initially incubated with the same concentrations of goat meat extract for 1 h, followed by LPS (Sigma-Aldrich) treatment at 1 μg/mL. For HaCaT cells, after exposure to goat meat extract at 50, 100, and 200 µg/mL, TNFα (Sigma-Aldrich) and IFNγ (Sigma-Aldrich) were applied at 10 ng/mL each. The negative control groups consisted of RAW264.7 cells without LPS treatment and HaCaT cells without TNFα and IFNγ exposure. The positive control groups included RAW264.7 cells treated with LPS to induce inflammation and HaCaT cells treated with TNFα and IFNγ. These positive controls served as benchmarks for comparing the effects of goat meat extract at different concentrations.

### 2.3. In Vitro Scratch Assay

HaCaT cells were plated in 12-well plates at a density of 1 × 10^5^ cells per well and cultured until reaching full confluence, followed by incubation in serum-free DMEM. A scratch wound was made in the cell monolayer using a sterile 200 µL pipette tip. The cells were rinsed with phosphate-buffered saline (PBS) and treated with goat meat extract at concentrations of 50, 100, and 200 µg/mL in fresh DMEM supplemented with 10% FBS. The treatment was performed for 0, 8, and 24 h at 37 °C in a 5% CO_2_ incubator. Wound width at each time point was measured at three distinct positions for each treatment group and compared to the initial width at 0 h. The wound width was expressed as a percentage relative to the untreated control group. Images were visualized, and wound width measurements were conducted using the Optiview program (Korea Lab Tech, Seongnam, Republic of Korea) to ensure accurate quantification of the wound healing process.

### 2.4. RNA Extraction and Quantitative Reverse-Transcription Polymerase Chain Reaction (qRT-PCR)

Total RNA was extracted from HaCaT and RAW264.7 cells using TRIzol reagent (Sigma-Aldrich) in accordance with the manufacturer’s protocol. RNA purity was confirmed by an A260/A280 ratio ranging from 1.8 to 2.0. The isolated RNA was converted into cDNA utilizing the iScript cDNA synthesis kit (Bio-Rad, Hercules, CA, USA). For gene expression analysis, SYBR Green Supermix (Bio-Rad) was employed. The cDNA samples underwent an initial denaturation at 95 °C for 30 s, followed by 40 amplification cycles of 95 °C for 5 s, 60 °C for 30 s, and final extension at 95 °C for 15 s using the 7500 Real-time PCR System (Applied Biosystems, Foster City, CA, USA). The threshold cycle (Cq) values were normalized to GAPDH as the reference gene to facilitate relative quantification.

### 2.5. Western Blot Analysis

The protein concentration in the cell lysates was measured using the Bradford reagent (Bio-Rad), with bovine serum albumin (Sigma-Aldrich) serving as the standard. Equal amounts of protein were heated at 95 °C for 5 min and resolved on an 8–12% SDS polyacrylamide gel. Proteins were subsequently transferred onto a nitrocellulose membrane and blocked for 1 h in PBS containing 0.1% Tween 20 and 5% skim milk. The membrane was then incubated at 4 °C overnight with primary antibodies specific to filaggrin (Abcam, Waltham, MA, USA), loricrin (Proteintech, Rosemont, IL, USA), involucrin (Proteintech), HAS1 (LSBio, Shirley, MA, USA), HAS2 (MYbiosource, San Diego, CA, USA), HAS3 (MYbiosource), JNK (Cell signaling, Danvers, MA, USA), p-JNK (Cell signaling), p38 (Cell signaling), p-p38 (Cell signaling), ERK (Invitrogen Inc., Carlsbad, CA, USA), p-ERK (Invitrogen), and β-actin (Sigma-Aldrich). After washing, secondary antibodies conjugated with peroxidase were applied to the membrane for 30 min. Following additional washes, the protein bands were visualized using an enhanced chemiluminescence substrate. Images were acquired with the ChemiDoc system (ThermoFisher Scientific, Waltham, MA, USA) and protein expression was quantified using the iBright analysis software5.3 (ThermoFisher Scientific).

### 2.6. Nitric Oxide (NO) Measurement

The nitrite concentration in the culture medium was measured as an indicator of NO production using the Griess assay (Promega, Madison, WI, USA). RAW264.7 cells were seeded at 5 × 10^4^ cells per well in 24-well plates and cultured for 24 h. Cells were pretreated with goat meat extract at concentrations of 50, 100, and 200 µg/mL for 1 h, followed by LPS (1 μg/mL) stimulation for 20 h. Afterward, equal volumes of culture medium and Griess reagent were mixed and incubated at room temperature for 10 min. The absorbance was then measured at 540 nm to quantify the nitrite levels.

### 2.7. Enzyme-Linked Immunosorbent Assay (ELISA)

HaCaT and RAW264.7 cells were cultured at a density of 5 × 10^4^ cells per well in 24-well plates, with varying concentrations (50, 100, and 200 µg/mL) of goat meat extract. Additionally, HaCaT cells were treated with TNFα and IFNγ at a concentration of 10 ng/mL each, while RAW264.7 cells were exposed to LPS at 1 µg/mL. After the treatments, the cell culture medium was centrifuged at 2000× *g* for 10 min to collect the supernatant. These supernatants were then analyzed for TNFα (Elabscience, Houston, TX, USA), IL-1 (Elabscience), IL-6 (Elabscience), CTSS (MYbiosource), MDC/CCL22 (MYbiosource), RANTES/CCL5 (MYbiosource), TARC/CCL17 (MYbiosource), filaggrin (MYbiosource), aquaporin (MYbiosource), and hyaluronic acid (Elabscience) using ELISA kits, following the manufacturer’s instructions.

### 2.8. Statistical Analysis

Data are presented as mean ± standard deviation (SD). For the statistical analysis, we utilized Student’s *t*-test and one-way analysis of variance (ANOVA), with statistical significance determined at *p* < 0.05.

## 3. Results and Discussion

### 3.1. Goat Meat Extract Enhances Keratinocyte Migration and Proliferation In Vitro

The ability of the goat meat extract to promote wound healing was evaluated through an in vitro scratch assay. The extract significantly accelerated wound closure in HaCaT cell monolayers, as indicated by a reduction in the scratch gap (Figure 1a,b). Specifically, after 24 h, the wound width in the goat meat extract-treated group was 18% smaller than that of the control group compared to measurements taken at 0 h. This effect indicated enhanced cellular migration and proliferation, suggesting that goat meat extract has the potential to facilitate skin repair and regeneration.

### 3.2. Goat Meat Extract Boosts Expression of Key Skin Barrier Genes in HaCaT Cells

Given that effective wound healing is closely linked to restoration and reinforcement of the skin barrier [31,32], we investigated the expression of genes involved in skin barrier function. Skin barrier and differentiation markers were examined using qRT-PCR and Western blotting, respectively. Treatment with goat meat extract resulted in a significant upregulation of filaggrin, loricrin, and involucrin, which are vital for epidermal differentiation and barrier function (Figure 2a,d). The enhancement in skin barrier function was indicated by the upregulation of structural proteins like filaggrin and involucrin, along with improved integrity of tight junctions [5,33]. The increased expression of tight junction proteins, such as occludin, claudin 1, claudin 4, and tricellulin, indicates enhanced intercellular cohesion and barrier integrity (Figure 2b) [34]. Furthermore, the extract elevated the expression of HAS1, HAS2, and HAS3, indicating a potential enhancement in hyaluronic acid synthesis, which is crucial for maintaining skin hydration and elasticity (Figure 2c,d) [35,36]. These results suggest that goat meat extract could be beneficial for fortifying the skin against environmental damage and preventing transepidermal water loss.

### 3.3. Goat Meat Extract Enhances the Function of the Skin Barrier in HaCaT Cells

The production of barrier-related proteins was quantified by ELISA. Treatment with goat meat extract markedly increased filaggrin and aquaporin levels, both of which are essential for skin barrier maintenance and hydration (Figure 3a,b). However, hyaluronic acid levels did not show significant changes, suggesting that its regulation may involve additional compensatory mechanisms that are not directly influenced by goat meat extract. The observed upregulation of *HAS1*, *HAS2*, and *HAS3* following treatment with goat meat extract indicated the activation of biosynthetic pathways involved in hyaluronic acid production (Figure 2b). Despite the increased expression of these genes, the actual hyaluronic acid levels remained unchanged (Figure 3c). This discrepancy underscores the complexity of hyaluronic acid regulation and suggests that while goat meat extract can enhance the expression of key synthetic genes, further research is necessary to elucidate the downstream effects and identify potential bottlenecks in hyaluronic acid production. Several factors may contribute to this discrepancy. Post-transcriptional modifications can impact mRNA stability and translation efficiency, influencing hyaluronic acid synthesis. Moreover, an increase in hyaluronidase activity, which degrades hyaluronic acid, could counteract the effects of the enhanced HAS gene expression. Feedback mechanisms might also be involved, where heightened gene expression could trigger downregulation of hyaluronic acid production once a certain threshold is reached. Additionally, external factors, such as other signaling molecules or stress conditions, could unexpectedly affect hyaluronic acid levels. To clarify this discrepancy, future studies should aim to explore these regulatory pathways. Specifically, examining the interplay between HAS gene expression, post-transcriptional modifications, and hyaluronidase activity could offer valuable insights into the role of goat meat extract in hyaluronic acid biosynthesis. Although goat meat extract promotes the expression of critical genes involved in hyaluronic acid production, the unchanged levels of hyaluronic acid highlight the complex regulation of its synthesis, warranting further investigation. Our results indicate that goat meat extract stimulates the production of filaggrin and aquaporin in HaCaT cells, both of which are essential for maintaining skin hydration and barrier integrity. Filaggrin plays a key role in strengthening the skin barrier and facilitating the production of natural moisturizing factors. It aids the maturation of keratinocytes and forms a protective skin barrier, and the products formed after its breakdown improve the water retention capacity of the skin [5]. Therefore, increased filaggrin expression can reduce water loss and enhance protection against external irritants. Aquaporin facilitates the transport of water through cell membranes and is essential for maintaining the overall moisture balance of the skin. Increased aquaporin expression boosts the skin’s water content and improves skin elasticity and smoothness [37]. These findings suggest that goat meat extract can improve skin hydration and barrier function in several ways. Filaggrin and aquaporin function through different pathways to enhance skin health.

### 3.4. Goat Meat Extract Exhibits Anti-Inflammatory Effects in HaCaT Cells

Skin immune disorders, including AD and allergies, are characterized by heightened skin inflammation [1,4,38,39]. Keratinocytes are pivotal in the inflammatory immune response when activated by various stimuli [40]. We examined the anti-inflammatory effects of goat meat extract on HaCaT cells under inflammatory conditions induced by TNFα and IFNγ. To examine the effect of goat meat extract on the secretion of pro-inflammatory chemokines and cytokines, HaCaT cells were pretreated with the extract and then exposed to TNFα and IFNγ. Chemokines play a vital role in the immune response and inflammation. The extract notably decreased the release of various chemokines and pro-inflammatory cytokines, including CTSS, MDC/CCL22, TARC/CCL17, RANTES/CCL5, TNFα, and IL-1, in a concentration-dependent manner (Figure 4a–f). However, IL-6 levels remained unchanged (Figure 4g). The significant reduction in pro-inflammatory mediator levels suggests that goat meat extract effectively exerts anti-inflammatory effects. Cytokines and chemokines play crucial roles in mediating inflammatory responses in the skin. Elevated levels of these molecules are often associated with inflammatory skin conditions, leading to symptoms such as redness, swelling, and irritation. The observed reduction in cytokine and chemokine levels suggests that goat meat extract may mitigate these inflammatory processes. A decrease in these inflammatory mediators may complement the previously noted increase in filaggrin and aquaporin expression. While filaggrin and aquaporin enhance the skin barrier function and hydration, a reduction in cytokines and chemokines can contribute to a calmer and less reactive skin environment. This dual action of strengthening the skin barrier and reducing inflammation suggests that goat meat extract could be a valuable natural ingredient for improving overall skin health and resilience.

### 3.5. Goat Meat Extract Exhibits Anti-Inflammatory Effects in RAW264.7 Cells

In RAW264.7 macrophages activated by LPS, treatment with goat meat extract led to a significant reduction in the production of the inflammatory mediators TNF-alpha, IL-1, and NO (Figure 5a–c). However, no significant difference in IL-6 levels was observed between the control and goat meat extract-treated groups (Figure 5d). This lack of change in IL-6 production was consistent with the results observed in HaCaT cells. This suggests that goat meat extract did not affect certain inflammatory pathways, indicating its potential selective action on other cytokine and chemokine pathways. Therefore, even in the absence of changes in IL-6, the reduction in TNFα, IL-1, and NO levels demonstrates an anti-inflammatory effect of the extract.

### 3.6. Anti-Inflammatory Effects of Goat Meat Extract on Keratinocytes Are Mediated Through the MAPK Signaling Pathway

MAPKs are essential protein kinases involved in regulating various cellular processes in response to external stressors [41]. In particular, MAPK is activated by stimulation from TNFα and IFNγ, leading to the increased expression of JNK, p38, and ERK [42,43]. Our study demonstrates that goat meat extract effectively reduces MAPK activation in HaCaT cells. Western blot analysis revealed significant inhibition of JNK, p38, and ERK phosphorylation in the goat meat extract-treated cells, with noticeable differences compared to the control group (Figure 6a). Furthermore, to explore the specific MAPK signaling pathways involved in the modulation of TNFα and IFNγ-induced pro-inflammatory cytokine production, such as IL-1 and various chemokines (CTSS, MDC/CCL22, and TARC/CCL17), we used selective kinase inhibitors: SP600125 for JNK, SB203580 for p38, and PD98059 for ERK. As demonstrated in Figure 6b–e, treatment with goat meat extract in combination with these inhibitors notably reduced the production of pro-inflammatory cytokines and chemokines induced by TNFα and IFNγ. These results indicate that goat meat extract may significantly influence inflammatory responses by inhibiting the MAPK signaling pathway. The ability of goat meat extract to suppress MAPK activation highlights its potential therapeutic application in controlling inflammation-related conditions. Given the integral role of MAPK in mediating cellular responses to stress and inflammation, the use of natural extracts, such as goat meat, could provide a complementary strategy for the management of inflammatory diseases.

## 4. Conclusions

Our study is the first to demonstrate that goat meat exhibits anti-atopic properties by suppressing the expression of cytokines and chemokines induced by TNFα and IFNγ through the inhibition of the MAPK pathway in HaCaT cells. Additionally, the extract significantly reduced the expression of inflammatory mediators in RAW264.7 cells. Furthermore, it enhanced the production of filaggrin and aquaporin, both of which are essential for maintaining the physical barrier function of the skin. However, this study was limited to in vitro experiments, and further research is needed to explore the effects of goat meat extract in animal models. In vivo studies will provide valuable insights into the direct impact of the extract on AD. While more research is required, our results suggest that goat meat extract shows promise as a potential therapeutic agent for reducing inflammation and managing AD.

## Figures and Tables

**Figure 1 foods-13-03934-f001:**
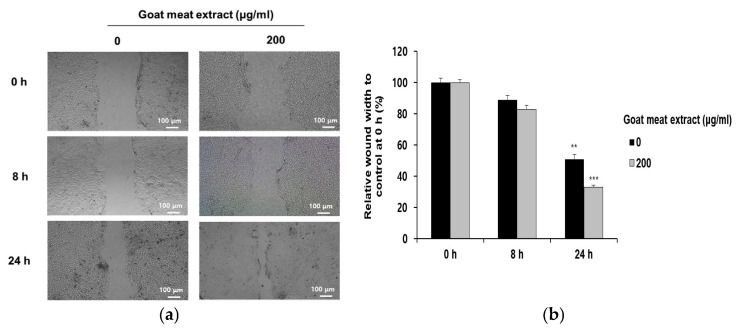
Effect of goat meat extract on HaCaT cell migration. (**a**) HaCaT cell migration was visualized and recorded using a 4× objective lens with a microscope camera at 0, 8, and 24 h post-scratching. (**b**) Cell migration was assessed using the wound healing assay. The data are presented as mean ± SD from three separate experiments. ** *p* < 0.01, *** *p* < 0.001.

**Figure 2 foods-13-03934-f002:**
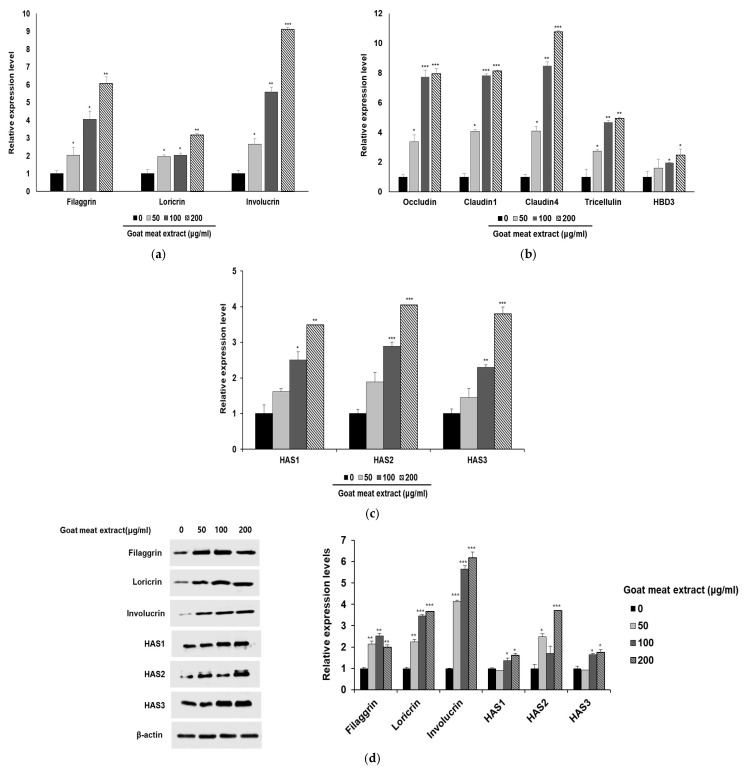
Effect of goat meat extract on the expression of genes related to skin barrier improvement. mRNA expression levels of (**a**) skin differentiation markers (filaggrin, loricrin, and involucrin) and (**b**) barrier function-related genes (occludin, claudin1, claudin 4, tricellulin, and HBD3) were determined via qRT-PCR analysis. (**c**) The mRNA expression of HAS1, HAS2, and HAS3 in HaCaT cells treated with goat meat extract was analyzed by qRT-PCR analysis. (**d**) Protein levels of filaggrin, loricrin, involucrin, HAS1, HAS2, and HAS3 were evaluated, with β-actin serving as a control. The data are presented as mean ± SD from three separate experiments. * *p* < 0.05, ** *p* < 0.01, *** *p* < 0.001.

**Figure 3 foods-13-03934-f003:**
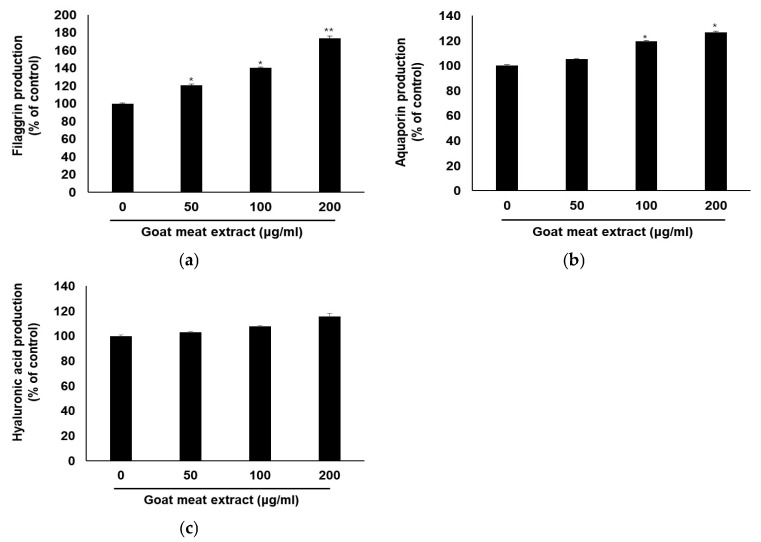
Impact of goat meat extract on physical barrier function and skin hydration. HaCaT cells were exposed to goat meat extract at concentrations of 50, 100, and 200 µg/mL for 24 h. Levels of (**a**) filaggrin, (**b**) aquaporin, and (**c**) hyaluronic acid were assessed using ELISA. The results are presented as mean ± SD from three independent experiments. * *p* < 0.05, ** *p* < 0.01.

**Figure 4 foods-13-03934-f004:**
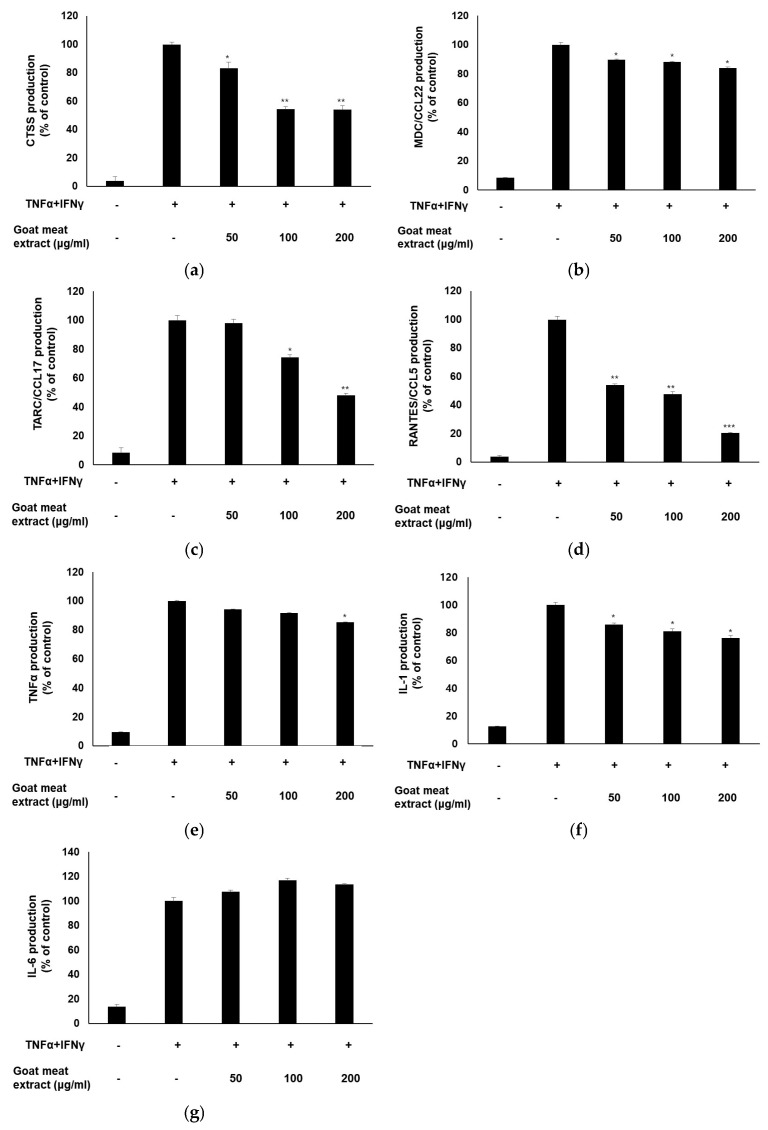
Effect of goat meat extract on the production of chemokines and pro-inflammatory cytokines. (**a**) CTSS, (**b**) MDC/CCL22, (**c**) TARC/CCL17, (**d**) RANTES/CCL5, (**e**) TNFα, (**f**) IL-1, and (**g**) IL-6 levels were measured using ELISA. The data are presented as mean ± SD from three separate experiments. * *p* < 0.05, ** *p* < 0.01, *** *p* < 0.001.

**Figure 5 foods-13-03934-f005:**
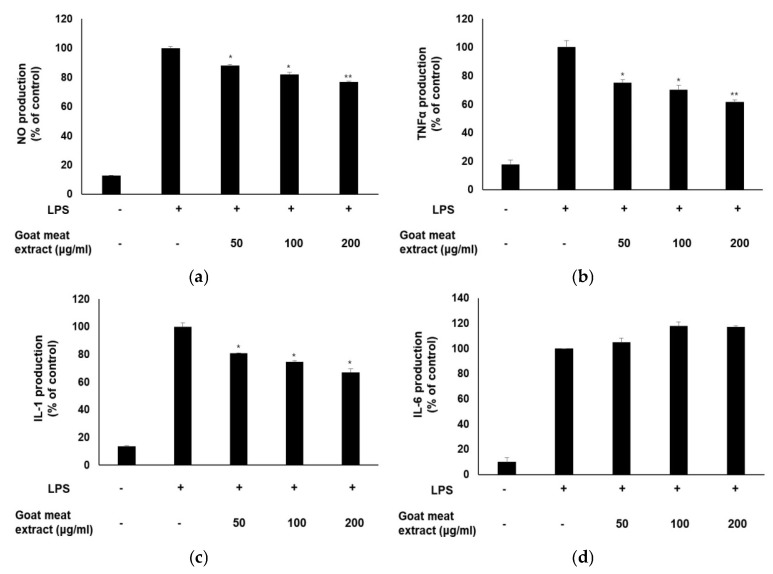
Impact of goat meat extract on pro-inflammatory cytokine production. (**a**) NO, (**b**) TNFα, (**c**) IL-1, and (**d**) IL-6 levels were quantified by ELISA. Data are expressed as mean ± SD from three independent experiments. * *p* < 0.05, ** *p* < 0.01.

**Figure 6 foods-13-03934-f006:**
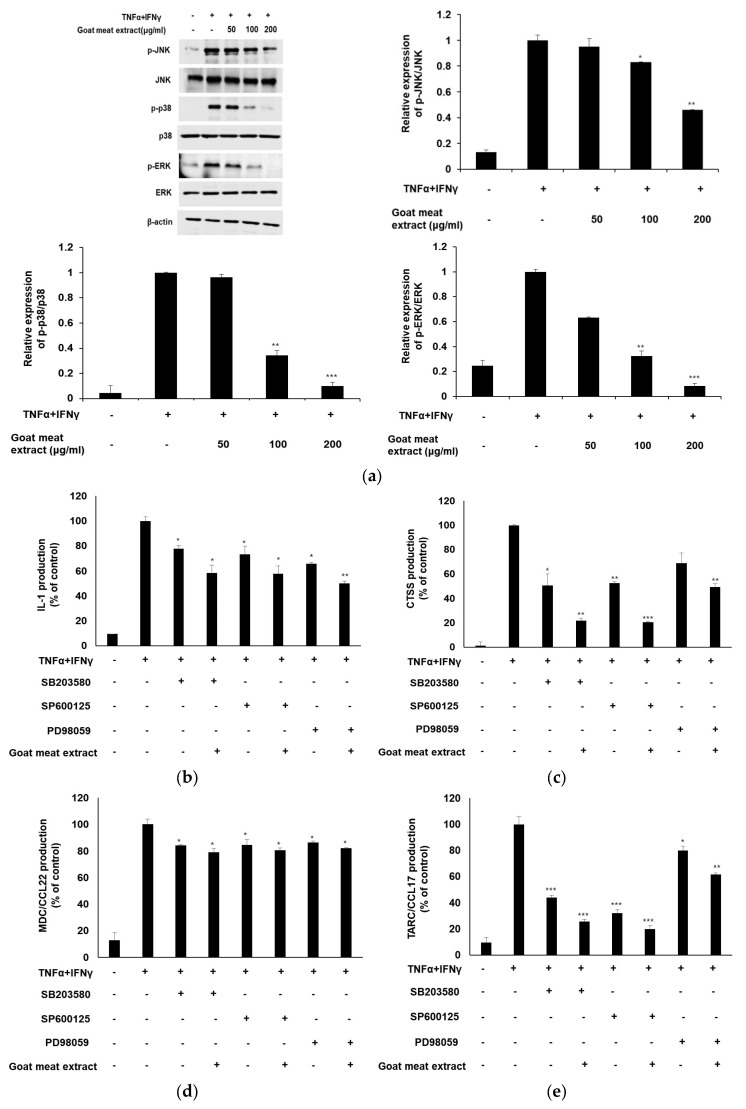
Effect of goat meat extract on TNFα and IFNγ induced MAPK signaling in HaCaT cells. Cells were pretreated with goat meat extract at concentrations of 50, 100, and 200 µg/mL for 1 h, followed by TNFα and IFNγ exposure for 1 h. (**a**) Western blot analysis was performed to assess the expression of p-JNK, JNK, p-p38, p38, p-ERK, and ERK. In cells treated with goat meat extract, the JNK inhibitor SP600125, p38 inhibitor SB203580, and ERK inhibitor PD98059 were applied. The levels of (**b**) IL-1, (**c**) CTSS, (**d**) MDC/CCL22, and (**e**) TARC/CCL17 were measured in HaCaT cells using ELISA. Data are expressed as mean ± SD from three independent experiments. * *p* < 0.05, ** *p* < 0.01, *** *p* < 0.001.

## Data Availability

The original contributions presented in this study are included in the article. Further inquiries can be directed to the corresponding author.

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
