# Peer review of "Skin Function Improvement and Anti-Inflammatory Effects of Goat Meat Extract"

_foods, 2024, doi:10.3390/foods13233934_

Round 1

Reviewer 1 Report

Comments and Suggestions for Authors

The article offers a compelling exploration of goat meat extracts as a potential therapeutic for enhancing skin function and reducing inflammation. Below are refined observations and recommendations for improvement:

a) The introduction provides a solid foundation by addressing the context of inflammatory skin diseases and the limitations of current treatments. However, the rationale for selecting goat meat could be strengthened by including a more detailed comparison with previously studied animal extracts. Establishing this connection more clearly would enhance the justification for the study.

b) Additional methodological details are needed, particularly regarding the specific concentrations of goat meat extract used, as well as the inclusion of negative and positive controls. This information is essential to improve reproducibility and the clarity of experimental design.

c) A typographical error was noted in the description of "qRT-PCR," where "revere-transcription" should be corrected to "reverse-transcription."

d) There is a notable inconsistency between the measured levels of hyaluronic acid (HA) and the expression levels of HAS1, HAS2, and HAS3.

While these genes showed increased expression, the levels of HA, as determined by ELISA, did not exhibit significant changes compared to the control. This discrepancy suggests that additional regulatory factors, such as post-transcriptional modifications or increased hyaluronidase activity, might be influencing HA production. A more in-depth discussion of these potential mechanisms is required to address this inconsistency. Please, improve the discussion.

e) The study's limitations, particularly the lack of in vivo models, should be critically examined. Including an acknowledgment of this limitation and discussing its implications for the study's conclusions would provide a more comprehensive perspective.

f) The conclusion is well-written but overly optimistic, given that the findings are based solely on in vitro experiments. Adjusting the tone to present a more balanced perspective would ensure that the conclusions align more closely with the study's scope and limitations.

Author Response

We would like to express our gratitude to the editor and all the reviewers for their insightful comments and suggestions, which have greatly enhanced the quality of our manuscript. We have revised the manuscript in accordance with the reviewer’s feedback, with changes highlighted in red throughout. Additionally, we have included a point-by-point response to each comment below. We sincerely hope that the revised manuscript meets the standards for publications.  

We greatly appreciate the valuable suggestions and comments from the Reviewer 1.

Point 1: The introduction provides a solid foundation by addressing the context of inflammatory skin diseases and the limitations of current treatments. However, the rationale for selecting goat meat could be strengthened by including a more detailed comparison with previously studied animal extracts. Establishing this connection more clearly would enhance the justification for the study.

Response 1: Thank you for your valuable feedback regarding the rationale for selecting goat meat. While we were unable to provide a specific comparison with previously studied animal extracts such as snail mucin or fish collagen, we have elaborated on the reasons for selecting goat meat, highlighting its unique nutritional and functional properties (page 2, line 15-23; restart line numbers on each page). We believe that these aspects make goat meat a valuable candidate for our study.

Point 2: Additional methodological details are needed, particularly regarding the specific concentrations of goat meat extract used, as well as the inclusion of negative and positive controls. This information is essential to improve reproducibility and the clarity of experimental design.

Response 2: Thank you for your valuable feedback. In response to your request for additional methodological details, we have now included the specific concentrations of goat meat extract utilized in our experiments, as well as the information regarding the inclusion of negative and positive controls (page 2, line 44- page 3, line 2; restart line numbers on each page). We believe that these additions will enhance the reproducibility and clarity of our experimental design.   

Point 3: A typographical error was noted in the description of “qRT-PCR”, where “revere-transcription” should be corrected to “reverse-transcription”

Response 3: Thank you for your feedback. I appreciate you pointing out the typographical error in the description of “qRT-PCR”. I have corrected “revere-transcription” to “reverse-transcription” (page 3, line 16; restart line numbers on each page). 

Point 4: There is a notable inconsistency between the measured levels of hyaluronic acid (HA) and the expression levels of HAS1, HAS2, and HAS3.

Response 4: Thank you for your constructive comments regarding the discrepancy between the expression levels of HAS genes and the measured levels of HA. We appreciate your feedback and have revised the discussion to address the points you raised (page 6, line 16-30; restart line numbers on each page). 

Point 5: The study’s limitations, particularly the lack of in vivo models, should be critically examined. Including an acknowledgment of this limitation and discussing its implications for the study’s conclusions would provide a more comprehensive perspective.

Response 5: Thank you for your valuable feedback regarding the limitations of our study, particularly concerning the lack of in vivo models. We appreciate your insights and have addressed this point in our revised conclusion raised (page 11, conclusion part).

Point 6: The conclusion is well-written but overly optimistic, given that the findings are based solely on in vitro experiments. Adjusting the tone to present a more balanced perspective would ensure that the conclusions align more closely with the study’s scope and limitations.

Response 6: Thank you for your insightful feedback regarding the conclusion of our study. We appreciate your perspective on the need for a more balanced tone, especially considering that our findings are based exclusively on in vitro experiments. In response to your suggestion, we have revised the conclusion to reflect a more cautious interpretation of our results. We emphasize the limitations of in vitro studies and clearly state that further research, particularly in vivo studies, is necessary to validate our findings (page 11, conclusion part).

Reviewer 2 Report

Comments and Suggestions for Authors

The work of Bae et al. is an interesting proposal, the authors propose that goat meat extract enhances skin function and reduces inflammation, making it a potential therapeutic agent for skin related conditions. Based on the above, some observations are made to improve the work:

 The structure and writing of the abstract are a bit repetitive and disorganized. For example, this could be improved by starting with background, objectives, methodology, results, and conclusion.

In the introduction, add more references to the known therapeutic effects of goat meat. Example: https://doi.org/10.1016/j.jff.2023.105410 ( Supplementation with goat meat extract improves exercise performance, reduces physiological fatigue, and modulates gut microbiota in mice)

In the methodology, add the ATCC reference of the cell lines used.

Add company and country of each reagent or kit used.

What was the number of cells used in each condition/assay?

What was the integrity of the extracted nucleic acids?

Preferably, use the same scale for the graphs, specifically review Figures 2 and 3.

Improve the quality of figures 1 and 2.

Figure 1a is missing the scale. Please indicate the microscope model and the program used to take the micrograph.

In general, check all abbreviations and spelling.

Comments on the Quality of English Language

Author Response

We would like to express our gratitude to the editor and all the reviewers for their insightful comments and suggestions, which have greatly enhanced the quality of our manuscript. We have revised the manuscript in accordance with the reviewer’s feedback, with changes highlighted in red throughout. Additionally, we have included a point-by-point response to each comment below. We sincerely hope that the revised manuscript meets the standards for publications.  

We greatly appreciate the valuable suggestions and comments from the Reviewer 2.

Point 1: The structure and writing of the abstract are a bit repetitive and disorganized.

Response 1: Thank you for your valuable feedback regarding the structure and organization of our abstract. We appreciate your suggestions for improvement. In response to your comments, we have revised the abstract to follow a clearer structure, beginning with background information, followed by objectives, methodology, results, and conclusion.

Point 2: In the introduction, add more references to the known therapeutic effects of goat meat.

Response 2: Thank you for your suggestion regarding the inclusion of more references to the therapeutic effects of goat meat in the introduction. In response to your feedback, we have added the study (page 2, line 24-29; restart line numbers on each page)

Point 3: In the methodology, add the ATCC reference of the cell lines used.

Response 3: Thank you for your helpful comments regarding the inclusion of the ATCC reference for the cell lines used in our study. In response to your feedback, we have added the relevant ATCC references to the methodology section to provide clearer information on the cell lines utilized (page 2, line 39-40; restart line numbers on each page).

Point 4: Add company and country of each reagent or kit used.  

Response 4: Thank you for your valuable suggestion regarding the inclusion of the company and country for each reagent or kit used in our study. In response to your feedback, we have added this information to the methodology section to provide more comprehensive details about the materials utilized.

Point 5: What was the number of cells used in each condition/assay?

Response 5: We appreciate your inquiry regarding the number of cells utilized in each experimental condition. We have added detailed information regarding the cell counts for each assay in the Materials and Methods section of the manuscript.

Point 6: What was the integrity of the extracted nucleic acids?

Response 6: Thank you for your inquiry regarding the integrity of the extracted nucleic acids. We have confirmed the purity of the RNA by measuring the A260/A280 ratio, which ranged from 1.8 to 2.0 (page 3, line 18-19; restart line numbers on each page). This indicates that the RNA samples were of high quality and suitable for downstream applications.  

Point 7: Preferably, use the same scale for the graphs, specifically review Figure 2 and 3.

Response 7: Thank you for your valuable feedback regarding the scale of the graphs. We have revised figures 2 and 3.

Point 8: Improve the quality of figures 1 and 2.

Response 8: Thank you for your feedback. In response, we have increased the resolution of the images and restructured the graphs to enhance clarity and visual quality. We believe these improvements will provide a better representation of the data.

Point 9: Figure is missing the scale. Please indicate the microscope model and the program used to take the micrograph. 

Response 9: Thank you for your valuable feedback regarding the figure. I would like to clarify that I have included the scale bar in the revised figure, and the program used to take the micrograph is specified in the text (page 3, line 12-15; restart line numbers on each page).

Point 10: In general, check all abbreviations and spelling.

Response 10: Thank you for your valuable feedback regarding the abbreviations and spelling in our manuscript. We have carefully reviewed the document and made the necessary corrections to ensure accuracy and clarity.